# Evidence Linking Cadmium Exposure and β_2_-Microglobulin to Increased Risk of Hypertension in Diabetes Type 2

**DOI:** 10.3390/toxics11060516

**Published:** 2023-06-08

**Authors:** Supabhorn Yimthiang, Phisit Pouyfung, Tanaporn Khamphaya, David A. Vesey, Glenda C. Gobe, Soisungwan Satarug

**Affiliations:** 1Occupational Health and Safety, School of Public Health, Walailak University, Nakhon Si Thammarat 80160, Thailand; ksupapor@mail.wu.ac.th (S.Y.); phisit.po@mail.wu.ac.th (P.P.); tanaporn.kh@mail.wu.ac.th (T.K.); 2The Centre for Kidney Disease Research, Translational Research Institute, Brisbane 4102, Australia; david.vesey@health.qld.gov.au (D.A.V.); g.gobe@uq.edu.au (G.C.G.); 3Department of Kidney and Transplant Services, Princess Alexandra Hospital, Brisbane 4102, Australia; 4School of Biomedical Sciences, The University of Queensland, Brisbane 4072, Australia; 5NHMRC Centre of Research Excellence for CKD QLD, UQ Health Sciences, Royal Brisbane and Women’s Hospital, Brisbane 4029, Australia

**Keywords:** blood pressure, β_2_-microglobulin, cadmium, diabetes, hypertension

## Abstract

The most common causes of chronic kidney disease, diabetes, and hypertension are significant public health issues worldwide. Exposure to the heavy metal pollutant, cadmium (Cd), which is particularly damaging to the kidney, has been associated with both risk factors. Increased levels of urinary β_2_-microglobulin (β_2_M) have been used to signify Cd-induced kidney damage and circulating levels have been linked to blood pressure control. In this study we investigated the pressor effects of Cd and β_2_M in 88 diabetics and 88 non-diabetic controls, matched by age, gender and locality. The overall mean serum β_2_M was 5.98 mg/L, while mean blood Cd and Cd excretion normalized to creatinine clearance (C_cr_) as E_Cd_/C_cr_ were 0.59 µg/L and 0.0084 µg/L of filtrate (0.95 µg/g creatinine), respectively. The prevalence odds ratio for hypertension rose by 79% per every ten-fold increase in blood Cd concentration. In all subjects, systolic blood pressure (SBP) showed positive associations with age (β = 0.247), serum β_2_M (β = 0.230), and E_Cd_/C_cr_ (β = 0.167). In subgroup analysis, SBP showed a strong positive association with E_Cd_/C_cr_ (β = 0.303) only in the diabetic group. The covariate-adjusted mean SBP in the diabetics of the highest E_Cd_/C_cr_ tertile was 13.8 mmHg higher, compared to the lowest tertile (*p* = 0.027). An increase in SBP associated with Cd exposure was insignificant in non-diabetics. Thus, for the first time, we have demonstrated an independent effect of Cd and β_2_M on blood pressure, thereby implicating both Cd exposure and β_2_M in the development of hypertension, especially in diabetics.

## 1. Introduction

Hypertension is defined as systolic blood pressure ≥ 140 mmHg, or diastolic blood pressure ≥90 mmHg, and is both a cause and a consequence of chronic kidney disease (CKD) [1,2,3]. An increased risk of hypertension has been linked to environmental exposure to cadmium (Cd), a widespread metal pollutant, in the general population in many countries, including the U.S. [4,5], Korea [6], China [7,8], Canada [9], Japan [10], and Thailand [11]. A cross-sectional study of the U.S. population observed an increased risk of hypertension in Caucasian and Mexican-American women who had blood Cd levels ≥0.4 µg/L [4].

A prospective cohort study of American Indians has provided strong support for a causal link between Cd exposure and hypertension [12]. In this study, higher urinary Cd excretion at baseline was associated with higher rates of systolic and diastolic blood pressure (SBP, DBP) increase. There was a 10% increase in risk for hypertension with every ten-fold increase in urinary Cd [12]. It is noteworthy that Cd exposure increased the risk of hypertension independently of smoking.

Cd accumulates primarily in the proximal tubular epithelial cells of the kidneys, where the burden of Cd as µg/g kidney tissue weight increases with age [13,14,15]. Kidney disease associated with chronic Cd exposure is primarily due to proximal tubule cell damage. This results in a sustained decline in glomerular filtration rate (GFR) and tubular proteinuria, evident from an increased excretion of the low-molecular weight protein, β_2_-microglobulin (β_2_M) [16]. Thus, an increase in β_2_M excretion is often used to reflect the impact of Cd on tubular protein reabsorption.

Genome-wide association and experimental studies have revealed many novel biological roles of β_2_M that include cardiovascular disease and blood pressure regulation [17,18,19]. In the Framingham Heart Study (*n* = 7065), an increased level of plasma β_2_M was linked to increased risks of prevalent and incident hypertension [19]. An increased level of urinary β_2_M was associated with an increased risk of hypertension in the Japanese general population [20]. Furthermore, an increased level of urinary β_2_M was associated with a 79% increased risk of a large decline in eGFR (10 mL/min/1.73 m^2^) over a five-year observation period [21].

The present study aimed to test the hypothesis that Cd exposure increases blood pressure, which in turn promotes the progression of diabetic kidney disease (DKD). Thus, we used a case-control design to investigate the relationships between serum β_2_M and blood Cd with SBP, DBP and hypertension according to the level of urinary Cd excretion, which reflects kidney burden. Excretion rates of Cd and β_2_M (E_Cd_ and E_β2M_) were normalized to creatinine clearance (C_cr_) as E_Cd_/C_cr_ and E_β2M_/C_cr_, respectively [22,23]. This C_cr_-normalization of an excretion rate depicts an amount of Cd or β_2_M excreted per volume of filtrate, which is at least roughly related to the amount of the chemical excreted per nephron. A C_cr_-normalized excretion rate is unaffected by creatinine excretion, while it corrects for differences in the number of surviving nephrons among study subjects [24]. Thus, E_Cd_/C_cr_ provides an accurate quantification of the kidney burden of Cd and its kidney effects.

## 2. Materials and Methods

### 2.1. Recruitment of Study Subjects

Diabetic cases together with non-diabetic controls matched by age, gender and locality were recruited from the health promotion center of Pakpoon Municipality, Nakhon Si Thammarat Province, Thailand. Previous studies suggest that the adverse effects of Cd, especially in kidneys, are more prevalent and more severe in women than men [25,26]. Thus, more women (80.7%) were recruited to maximize the likelihood of finding an effect of Cd, when the sample size was modest (*n* = 176). The inclusion criteria were residents in the Pakpoon municipality, 40 years of age or older who were diagnosed with type 2 diabetes or were apparently healthy. The exclusion criteria were residents in other municipalities, pregnancy, breast-feeding, hospital records or physician’s diagnosis of an advanced chronic disease. All subjects were provided with details of study objectives, study procedures, benefits, and potential risks, and they all provided their written informed consent prior to participation.

The sociodemographic data, education attainment, occupation, health status, family history of diabetes, and smoking status were obtained by structured interview questionnaires. Prediabetes and diabetes were indicated by fasting plasma glucose levels ≥110 and ≥126 mg/dL, respectively (https://www.cdc.gov/diabetes/basics/getting-tested.html (accessed on 5 May 2023). Hypertension was defined as measured systolic blood pressure ≥140 mmHg, or diastolic blood pressure ≥90 mmHg [3]. After excluding subjects with missing data, 176 subjects (88 diabetics and 88 apparently healthy, non-diabetic controls) were included in this study.

### 2.2. Blood and Urine Sampling and Analysis

Participants were requested to fast overnight, and the collection of blood and urine samples was carried out at a local health center of Pakpoon Municipality on the morning of the following day. For glucose assay, blood samples were collected in tubes containing fluoride that inhibited glycolysis. Blood samples for Cd analysis were collected in separate tubes containing ethylene diamine tetra-acetic acid (EDTA) as an anticoagulant. The blood and urine samples were kept on ice and transported within 1 h to the laboratory of Walailak University, where plasma samples were prepared. Aliquots of urine, whole blood and plasma samples were stored at −80 °C for later analysis.

Fasting plasma glucose concentrations ([Glc]_p_) were measured to ascertain diabetes diagnosis and diabetes free stage of controls. The assay of the plasma concentration of glucose was based on colorimetry. Assays of creatinine in urine and plasma ([cr]_u_, [cr]_p_]) were based on the Jaffe reaction. The urine concentration of albumin ([Alb]_u_) was determined using an immunoturbidimetric method. The human beta-2 microglobulin/β_2_M ELISA pair set (Sino Biological Inc., Wayne, PA, USA) was employed to determine serum and urine concentration of β_2_M ([β_2_M]_s_ and [β_2_M]_u_) with the low detection limit of 3.13 pg/mL.

### 2.3. Determinarion of Blood and Urinary Concentration of Cadmium

Blood Cd concentration ([Cd]_b_) was determined with the atomic absorption spectrophotometer (GBC Scientific Equipment, Hampshire, IL, USA). Multielement standards were used to calibrate metal analysis (Merck KGaA, Darmstadt, Germany). For the purposes of quality control, analytical accuracy, and precision assurance, reference urine and whole blood metal control levels 1, 2, and 3 (Lyphocheck, Bio-Rad, Hercules, CA, USA) were used. All test tubes, bottles, and pipettes used in the metal analysis were acid-washed and rinsed thoroughly with deionized water. When a [Cd]_b_ level was less than its detection limits, the concentration assign was the detection limit divided by the square root of 2 [27]. Sixty-one subjects (34.6%) had [Cd]_b_ below the detection limit of 0.1 µg/L.

### 2.4. Normalization of the Excretion of Cadmium and β_2_Microglobulin

E_x_ was normalized to E_cr_ as [x]_u_/[cr]_u_, where x = Cd or β_2_M; [x]_u_ = urine concentration of x (mass/volume); and [cr]_u_ = urine creatinine concentration (mg/dL). The ratio [x]_u_/[cr]_u_ was expressed in μg/g of creatinine.

E_x_ was normalized to C_cr_ as E_x_/C_cr_ = [x]_u_[cr]_p_/[cr]_u_, where x = Cd or β_2_M; [x]_u_ = urine concentration of x (mass/volume); [cr]_p_ = plasma creatinine concentration (mg/dL); and [cr]_u_ = urine creatinine concentration (mg/dL). E_x_/C_cr_ was expressed as the excretion of x per volume of filtrate [24].

### 2.5. Computation of the Estimated Glomerular Filtration Rate

The GFR is the product of nephron number and mean single nephron GFR, and in theory, the GFR is indicative of nephron function [28,29,30]. In practice, the GFR is estimated from established chronic kidney disease–epidemiology collaboration (CKD-EPI) equations and is reported as eGFR [31,32].

Male eGFR = 141 × [plasma creatinine/0.9]^Y^ × 0.993^age^,
where Y = −0.411 if [cr]_p_ ≤ 0.9 mg/dL, and Y = −1.209 if [cr]_p_ > 0.9 mg/dL.

Female eGFR = 144 × [plasma creatinine/0.7]^Y^ × 0.993^age^,
where Y = −0.329 if [cr]_p_ ≤ 0.7 mg/dL, and Y = −1.209 if [cr]_p_ > 0.7 mg/dL.

### 2.6. Statistical Analysis

Data were analyzed with IBM SPSS Statistics 21 (IBM Inc., New York, NY, USA). The Mann–Whitney U-test was used to assess differences in means between the two groups, and the Pearson chi-squared test was used to assess differences in percentages. Before data were subjected to parametric statistical analysis, the one-sample Kolmogorov–Smirnov test was used to identify a departure from a normal distribution of any continuous variables, and logarithmic transformation was applied when the variables showed rightward skewing. The multivariable logistic regression analysis was used to determine the Prevalence Odds Ratio (POR) for categorical outcomes. Obesity was designated when BMI > 30 kg/m^2^. Reduced eGFR was assigned when eGFR ≤ 60 mL/min/1.73 m^2^.

We employed the univariate analysis of covariance with Bonferroni correction in multiple comparisons to obtain the mean SBP, mean DBP and mean β_2_M excretion, adjusted for covariates and interactions. For all tests, *p*-values ≤ 0.05 were considered to indicate statistical significance.

## 3. Results

### 3.1. Study Subjects

Among 176 participants, 88 were diagnosed with diabetes, and 88 were apparently healthy controls who did not have diabetes (Table 1).

Women constituted 80% of the diabetics and controls. The mean diabetes duration was 9.3 years, and the overall mean age was 60 years. The overall percentages (%) of smokers and the obese were 9.7% and 10.8%. Obesity was 2.78-times more prevalent in diabetics, while % smoking, hypertension, and reduced eGFR did not differ. The mean BMI, mean SBP, mean serum β_2_M, mean urine β_2_M, mean E_β2M_/C_cr_, and mean E_β2M_/E_cr_ all were higher in the diabetics than controls. In contrast, mean eGFR, mean blood Cd, mean urine Cd, mean E_Cd_/C_cr_, and mean E_Cd_/E_cr_ were similar in the two groups.

### 3.2. Urinary Cd and Serum β_2_M as the Predictors of Blood Pressure Measures 

The associations of serum β_2_M and/or Cd exposure with SBP and DBP were evaluated with the regression analysis, incorporating age, BMI, serum β_2_M, Cd excretion rate, gender and smoking as the independent variables (Table 2).

In all subjects, age, BMI, serum β_2_M, urinary Cd excretion, gender and smoking altogether contributed to 12.3% of the total variation of SBP (*p* = 0.001). Of these six variables, SBP showed a moderate association with age (β = 0.247), serum β2M (β = 0.230), urinary Cd excretion (β = 0.167), and smoking (β = −0.218). In subgroup analysis, SBP was associated with urinary Cd excretion only in the diabetics (β = 0.303), while it showed a strong association with age only in non-diabetics (β = 0.348).

In the equivalent DBP regressions, Cd excretion was associated with DBP only when all subjects were included (β = 0.167), and this DBP-Cd association became insignificant in the subgroup analysis (Table 3).

Additional multiple linear regressions of blood pressure were conducted to evaluate gender differences in the effect of Cd on blood pressure (Table 4).

In men, SBP was associated strongly with both serum β_2_M (β = 0.415) and urinary Cd excretion (β = 0.432), while DBP was strongly associated only with urinary Cd excretion (β = 0.454). The association of the DBP in men with serum β_2_M was not significant (β = 0.242, *p* = 0.212). In comparison, there was no significant association between SBP and urinary Cd excretion in women. There was, however, a moderate association of SBP with age (β = 0.310) and serum β_2_M (β = 0.225).

### 3.3. Age and Serum β_2_M as Predictors of eGFR Decline

Table 5 shows the results of the eGFR regression analysis that assessed the adverse effect of an increased level of serum β_2_M on kidney outcome in relation to other variables.

Age, BMI, eGFR, serum β_2_M, excretion rates of Cd, smoking, gender, and hypertension together contributed to 19.1% and 15.8% of the variability of eGFR in all subjects (*p* < 0.001), and women (*p* = 0.001), respectively. In contrast, the variation in eGFR among men was not significantly associated with any of these independent variables.

eGFR was inversely associated with age (β = −0.307) and serum β_2_M (β = −0.235) when all subjects were included. In subgroup analysis, the inverse association of eGFR with age in women was maintained, but the association of eGFR and serum β2M was weakened and became statistically insignificant. 

To evaluate the dose-effect relationship between Cd exposure and blood pressure increase, we compared mean SBP and mean DBP values for non-diabetics and diabetics grouped according to E_Cd_/C_cr_ tertiles (Figure 1).

A linear increase in SBP with urinary Cd excretion rate was evident in the diabetic group only (Figure 1a). The covariate-adjusted mean SBP in the diabetics of the low, middle and high E_Cd_/C_cr_ tertiles were 133, 140 and 147 mmHg, respectively (*F* = 3.73, η^2^ 0.123, *p* = 0.031). The covariate-adjusted mean SBP tertile was 13.8 mmHg higher in the diabetics of the high than those of the lowest tertile (*p* = 0.027) (Figure 1b). The cutoff values of (E_Cd_/C_cr_) × 100 for tertiles 1, 2 and 3 were ≤0.069, 0.070–0.177 and ≥0.178 µg/L filtrate, respectively.

In comparison, the relationship between DBP and urinary Cd excretion was insignificant in both the diabetic and non-diabetic groups (Figure 1c). The covariate-adjusted mean DBP was similar across E_Cd_/C_cr_ tertile groups (Figure 1d).

### 3.4. An Association of Blood Cadmium and Increased Risk of Hypertension

Table 6 provides the results of the logistic regression analysis of hypertension that incorporated seven independent variables; age, BMI, blood Cd, serum β_2_M, gender, smoking, and diabetes.

Age, BMI, serum β_2_M, gender, smoking and diabetes did not show a significant association with hypertension, but BMI and blood Cd did. BMI was associated with an increase in POR for hypertension by 8% with every 1 kg/m^2^ increase in BMI (*p* = 0.042). Blood Cd as log_10_ ([Cd]_b_) × 10^3^) was associated with a 1.72-fold increase in POR for hypertension (*p* = 0.013).

### 3.5. Determinants of Blood Cadmium in Men and Women

Table 7 provides the results of blood Cd regression analysis of that incorporated ten independent variables; age, BMI, serum β_2_M, urinary excretion rates of β_2_M and Cd, gender, smoking, diabetes and hypertension.

All 10 independent variables contributed to 19.7% and 19.3% of the variability of blood Cd in all subjects (*p* < 0.001) and women (*p* < 0.001), respectively.

Blood Cd was not associated with serum β_2_M or the excretion rate of β_2_M, but this parameter was strongly associated with Cd excretion rate, especially among women (β = 0.394). In addition, blood Cd showed a moderate association with hypertension in women only (β = 0.181). In contrast, blood Cd in men was not associated with Cd excretion rate or hypertension, but it did show a strong positive association with smoking (β = 0.496).

## 4. Discussion

One of the most widely recognized consequences of chronic kidney damage is hypertension. Hypertension originating from kidney disease is associated predominately with the renin-angiotensin system of blood pressure control [33,34]. While hypertension due to frank renal pathology is relatively well understood, much milder kidney pathology may also cause more subtle, hitherto unsuspected, and earlier changes [35]. An increased level of β_2_M excretion, a common sign of Cd-induced nephrotoxicity, has been linked to an increased risk of hypertension in a Japanese population study [21]. In other studies, serum β_2_M predicted rapid decline in eGFR, diabetic kidney disease (DKD), cardiovascular disease, and mortality in diabetic patients [36,37,38]. However, the potential role of circulating β_2_M in the development of hypertension in diabetics who are also exposed to environmental Cd has never been investigated.

The higher mean serum β_2_M seen in diabetics compared with controls (7.03 vs. 4.93 mg/L, *p* = 0.002, Table 1) was consistent with other literature reports [37,38]. The independent association of this parameter (serum β_2_M) and urinary Cd with SBP was evident from regression analysis (Table 2). In all subjects, SBP showed a moderate association with serum β_2_M (β = 0.230) and Cd excretion rate (β = 0.167), while DBP was associated with excretion of Cd only (β =0.172). In subgroup analysis, SBP was associated with urinary Cd in the diabetic group only (β = 0.303), while an association of DBP with urinary Cd became insignificant due to a reduction in sample size (Table 3). The association of SBP with serum β_2_M was also weakened and became insignificant in subgroup analysis for the same reason as the association of DBP and urinary Cd excretion (Table 2 and Table 3).

The gender difference was seen in the association of β_2_M and Cd excretion with blood pressure (Table 4). Among men, SBP was strongly associated with both serum β_2_M (β = 0.415) and Cd excretion (β = 0.432), while DBP was strongly associated only with Cd excretion (β = 0.454). Our finding may help explain the results of a recent Danish cohort study of non-smokers which found an association between environmental Cd exposure and incident heart failure, especially among men [39].

Another gender difference was indicated by the regression analysis (Table 5), where eGFR showed a tendency to be inversely associated with serum β_2_M particularly in women. This observation is in line with previous studies showing serum β_2_M as the predictor of rapid decline in eGFR and DKD [36,37,38].

A dose-effect relationship between SBP and urinary Cd excretion in the diabetic group was apparent in the univariate/covariance analysis (Figure 1b), where urinary Cd excretion explained 12.3% of the variation in SBP (*F* = 3.73, *p* = 0.031) after adjustment for covariates and interactions. The covariate-adjusted mean SBP in those in the high E_Cd_/C_cr_ tertile was 13.8 mmHg higher than those in the low tertile (*p* = 0.027) (Figure 1b).

In the logistic regression analysis (Table 6), neither diabetes nor smoking increased the risk of hypertension significantly, but BMI and blood Cd did. The risk of hypertension rose by 8% and 79% with every 1-kg/m^2^ increase in BMI (*p* = 0.042), and ten-fold increase in blood Cd concentration (*p* = 0.013), respectively. Of note, blood Cd levels ≥ 0.4 µg/L were found to be associated with 1.5-fold to 2.4-fold increases in risk of hypertension among Caucasian and Mexican-American women enrolled in the U.S. National Health and Nutrition Examination Survey (NHANES) 1999–2006 (*n* = 16,222) [4]. Studies from other countries, including Korea [6], China [7,8], Canada [9] and Japan [10] have also linked Cd exposure indices (blood, serum or urinary Cd) to an increased risk of hypertension.

Cd exposure increased the risk of DKD in a Dutch cross-sectional study, including 231 patients with type 2 diabetes [40]. In a six-year median follow-up of these 231 diabetic patients, Cd exposure was associated with a progressive reduction of eGFR [41]. Thus, exposure to even low levels of environmental Cd promotes the development and progression of DKD. Data from the present study suggests that the kidney disease progression in diabetics who are also exposed to Cd could be attributed to the pressor effect of Cd.

Diabetes and hypertension are the major causes of CKD and subsequently development of kidney failure [42,43]. Collectively, our data indicate that environmental Cd exposure experienced by current populations in many countries has reached levels that adversely affect kidneys in a significant proportion, thereby arguing strongly for public measures to reduce exposure to Cd from all sources. Mitigation of Cd toxicity outcomes is equally necessary, given that therapeutically effective chelating agents to reduce kidney Cd burden are currently lacking. High rates of soil-to-plant transfer of Cd coupled with continuing mobilization of small amounts of the metal from non-bioavailable geologic matrices into biologically accessible situations predicts that human exposure to dietary Cd will continue to rise as will kidney failure.

## 5. Conclusions

This study shows that environmental exposure to low levels of Cd adversely affects blood pressure and GFR in both non-diabetic and diabetic. For the first time, an independent effect of Cd and β_2_M on SBP and eGFR has been demonstrated. This implies that both Cd exposure and circulating β_2_M are involved in the development of hypertension, and eGFR decline, especially in diabetics.

## Figures and Tables

**Figure 1 toxics-11-00516-f001:**
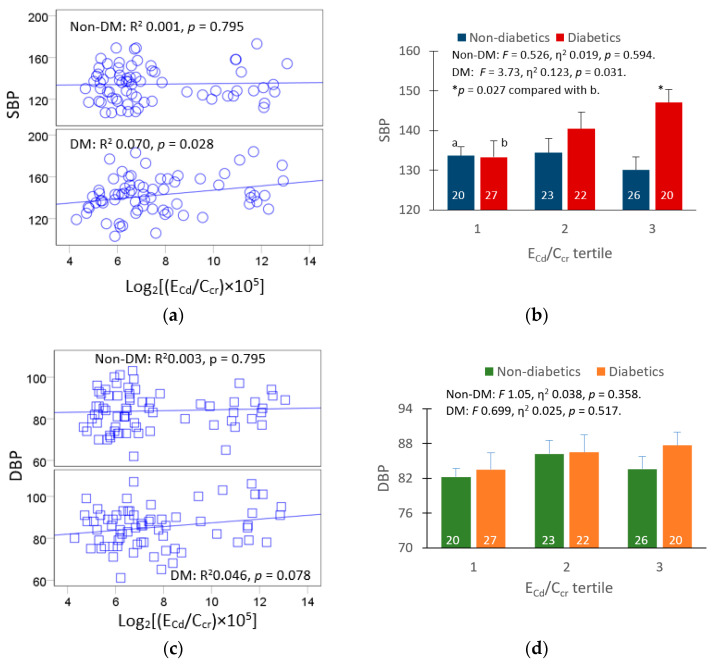
Dose-effect relationship of cadmium exposure and blood pressure. Scatterplots relates SBP (**a**) and DBP (**c**) to log[(E_Cd_/C_cr_) × 10^5^] in the non-diabetics and diabetics. Bar graphs depict mean SBP (**b**) and mean DBP (**d**) for non-diabetic and the diabetic subsets across E_Cd_/C_cr_ tertiles. Coefficients of determination (R^2^) and *p*-values are provided for all scatterplots. The numbers of subjects are provided for all subgroups. The letters a and b denote the non-diabetics and diabetics with the lowest tertile of (E_Cd_/C_cr_) × 100, respectively. Mean SBP and mean DBP values all were adjusted for covariates and interactions. Mean (SD) values for (E_Cd_/C_cr_) × 100 tertiles 1, 2 and 3 are 0.046 (0.014), 0.109 (0.030), and 2.316 (2.197), µg/L of filtrate, respectively.

**Table 1 toxics-11-00516-t001:** Characteristics of Study Subjects.

Parameters	All Subjects, *n* = 176	Non-Diabetics, *n* = 88	Diabetics, *n* = 88	*p*
Age, years	59.9 ± 9.7	60.4 ± 9.2	59.3 ± 10.2	0.389
Duration of diabetes, years	n/a	0	9.3 ± 7.6	−
Fasting plasma glucose, mg/dL	132 ± 61	94 ± 12	169 ± 68	<0.001
BMI, kg/m^2^	25.4 ± 4.7	24.7 ± 4.4	26.1 ± 5.0	0.024
Obese ^a^, %	10.8	5.7	15.9	0.029
Female, %	80.7	80.7	80.7	1.0
Smoker, %	9.7	11.4	8.0	0.444
Systolic blood pressure	138 ± 17	135 ± 17	141 ± 17	0.015
Diastolic blood pressure	84 ± 9	84 ± 9	83 ± 10	0.515
Hypertension, %	52.0	44.7	59.1	0.058
eGFR ^b^, mL/min/1.73 m^2^	79.4 ± 18.0	79.4 ± 14.4	79.5 ± 21.2	0.519
Reduced eGFR ^c^, %	16.5	11.4	21.6	0.067
Plasma creatinine, mg/dL	0.87 ± 0.24	0.85 ± 0.16	0.89 ± 0.30	0.834
Urine creatinine, mg/dL	89.2 ± 54.1	97.4 ± 52.6	81.0 ± 54.6	0.012
Serum β_2_M, mg/L	5.98 ± 3.41	4.93 ± 2.22	7.03 ± 4.03	0.002
Urine β_2_M, µg/L	740 ± 696	414 ± 362	1071 ± 793	<0.001
Blood Cd, µg/L	0.59 ± 0.74	0.64 ± 0.85	0.53 ± 0.60	0.986
Urine Cd, µg/L	0.68 ± 1.18	0.66 ± 1.07	0.70 ± 1.29	0.862
Normalized to C_cr_ (E_x_/C_cr_) ^d^				
(E_Cd_/C_cr_) × 100, µg/L filtrate	0.84 ± 1.66	0.86 ± 1.69	0.82 ± 1.64	0.389
(E_β2M_/C_cr_) × 100, µg/L filtrate	1313 ± 2397	543 ± 625	2104 ± 3175	<0.001
Normalized to E_cr_ (E_x_/E_cr_) ^e^				
E_Cd_/E_cr_, µg/g creatinine	0.96 ± 1.83	0.99 ± 1.94	0.92 ± 1.73	0.482
E_β2M_/E_cr_, µg/g creatinine	1284 ± 1747	633 ± 762	1954 ± 2178	<0.001

*n*, number of subjects; BMI, body mass index; β_2_M, β_2_-microglobulin; eGFR, estimated glomerular filtration rate; E_x_, excretion of x; cr, creatinine; C_cr_, creatinine clearance; Cd, cadmium; ^a^ Obese was defined as BMI >30 kg/m^2^; ^b^ eGFR, was determined by CKD-EPI equations [31]; ^c^ reduced eGFR corresponds to eGFR ≤60 mL/min/1.73 m^2^; ^d^ E_x_/E_cr_ = [x]_u_/[cr]_u_; ^e^ E_x_/C_cr_ = [x]_u_[cr]_p_/[cr]_u_, where x = β_2_M or Cd [24]. Data for all continuous variables are arithmetic means ± standard deviation (SD). For all tests, *p* ≤ 0.05 identifies statistical significance, determined by Pearson chi-square test for % differences and by the Mann–Whitney U-test for mean differences between diabetic and non-diabetic groups.

**Table 2 toxics-11-00516-t002:** Predictors of systolic blood pressure in controls and cases.

Independent Variables/Factors	SBP, mmHg					
	All Subjects		Non-Diabetics		Diabetics	
	β	*p*	β	*p*	β	*p*
Age, years	0.247	0.006	0.348	0.005	0.238	0.081
BMI, kg/m^2^	0.124	0.149	0.140	0.257	−0.016	0.901
Log_10_ ([β_2_M]_s_) × 10^3^), mg/L	0.230	0.006	0.209	0.082	0.153	0.235
Log_2_[(E_Cd_/C_cr_) × 10^5^], µg/L	0.167	0.042	−0.009	0.939	0.303	0.012
Gender	−0.119	0.251	−0.080	0.598	−0.094	0.526
Smoking	−0.218	0.033	−0.263	0.072	−0.114	0.446
Adjusted R^2^	0.123	0.001	0.152	0.011	0.089	0.065

β, standardized regression coefficient; adjusted R^2^, coefficient of determination. Coding, male 1, female 2; non-smoker 1, smoker 2. β indicates strength of association of SBP or DBP with independent variables (first column). Adjusted R^2^ indicates a fractional variation of SBP or DBP explained by all independent variables. For each test, *p*-values ≤ 0.05 indicate a statistically significant contribution of an independent variable to SBP or DBP variability.

**Table 3 toxics-11-00516-t003:** Predictors of diastolic blood pressure in controls and cases.

Independent Variables/Factors	DBP, mmHg					
	All Subjects		Non-Diabetics		Diabetics	
	β	*p*	β	*p*	β	*p*
Age, years	−0.110	0.233	−0.086	0.520	−0.137	0.323
BMI, kg/m^2^	0.041	0.645	0.112	0.406	−0.072	0.581
Log_10_ ([β_2_M]_s_) × 10^3^), mg/L	0.132	0.130	0.072	0.580	0.155	0.240
Log_2_[(E_Cd_/C_cr_) × 10^5^], µg/L	0.172	0.045	0.117	0.378	0.223	0.067
Gender	−0.213	0.052	−0.142	0.391	−0.255	0.096
Smoking	−0.207	0.053	−0.276	0.085	−0.110	0.475
Adjusted R^2^	0.042	0.069	−0.022	0.451	0.044	0.186

β, standardized regression coefficient; adjusted R^2^, coefficient of determination. Coding, male 1, female 2; non-smoker 1, smoker 2; code 1 is control. β indicates strength of association of SBP or DBP with independent variables (first column). Adjusted R^2^ indicates a fractional variation of SBP or DBP explained by all independent variables. For each test, *p*-values ≤ 0.05 indicate a statistically significant contribution of an independent variable to SBP or DBP variability.

**Table 4 toxics-11-00516-t004:** Comparing the predictors of systolic and diastolic blood pressure in men and women.

Independent Variables/Factors	SBP				DBP			
	Men		Women		Men		Women	
	β	*p*	β	*p*	β	*p*	β	*p*
Age, years	0.247	0.006	0.348	0.005	0.238	0.081	−0.110	0.233
BMI, kg/m^2^	0.124	0.149	0.140	0.257	−0.016	0.901	0.041	0.645
Log_10_ ([β_2_M]_s_) × 10^3^), mg/L	0.230	0.006	0.209	0.082	0.153	0.235	0.132	0.130
Log_2_[(E_Cd_/C_cr_) × 10^5^], µg/L	0.167	0.042	−0.009	0.939	0.303	0.012	0.172	0.045
Smoking	−0.218	0.033	−0.263	0.072	−0.114	0.446	−0.207	0.053
Adjusted R^2^	0.123	0.001	0.152	0.011	0.089	0.065	0.042	0.069

β, standardized regression coefficient; adjusted R^2^, coefficient of determination. Coding, non-smoker 1, smoker 2. β indicates strength of association of SBP or DBP with independent variables (first column). Adjusted R^2^ indicates a fractional variation of SBP or DBP explained by all independent variables. For each test, *p*-values ≤ 0.05 indicate a statistically significant contribution of an independent variable to SBP or DBP variability.

**Table 5 toxics-11-00516-t005:** Inverse associations of eGFR with age and serum β_2_M.

Independent Variables/Factors	eGFR, mL/min/1.73 m^2^					
	All Subjects		Men		Women	
	β	*p*	β	*p*	β	*p*
Age, years	−0.307	<0.001	−0.269	0.265	−0.299	0.002
BMI, kg/m^2^	0.038	0.651	−0.101	0.639	0.072	0.461
Log_10_ ([β_2_M]_s_) × 10^3^), mg/L	−0.235	0.004	−0.397	0.093	−0.176	0.059
Log_2_[(E_Cd_/C_cr_) × 10^5^], µg/L	−0.142	0.074	0.112	0.569	−0.173	0.061
Smoking	0.028	0.777	0.012	0.956	−0.034	0.712
Gender	0.083	0.408	−	−	−	−
Hypertension	0.035	0.662	0.00001	1.000	0.021	0.821
Adjusted R^2^	0.191	<0.001	0.150	0.132	0.158	0.001

β, standardized regression coefficient; adjusted R^2^, coefficient of determination. Coding, non-smoker 1, smoker 2; male 1, female 2; normotension 1, hypertension 2. β indicates strength of association eGFR with independent variables (first column). Adjusted R^2^ indicates a fractional variation of eGFR explained by all independent variables. For each test, *p*-values ≤ 0.05 indicate a statistically significant contribution of an independent variable to eGFR variability.

**Table 6 toxics-11-00516-t006:** Prevalence odds ratios for hypertension in relation to blood cadmium, serum β_2_M and other independent variables.

Independent Variables/ Factors	Hypertension				
	β Coefficients	POR	95% CI		*p*
	(SE)		Lower	Upper	
Age, years	0.019 (0.019)	1.019	0.982	1.058	0.318
BMI, kg/m^2^	0.081 (0.040)	1.084	1.003	1.172	0.042
Log_10_ ([Cd]_b_) × 10^3^), mg/L	0.544 (0.237)	1.723	1.083	2.741	0.022
Log_10_ ([β_2_M]_s_) × 10^3^), µg/L	0.374 (0.700)	1.454	0.369	5.732	0.593
Gender	0.653 (0.556)	1.922	0.646	5.719	0.240
Smoking	1.331 (0.738)	3.785	0.890	16.09	0.071
Non-DM	Referent				
<10-y DM	0.413 (0.047)	1.512	0.601	3.802	0.380
≥10-y DM	0.682 (0.0426)	1.977	0.857	4.560	0.110

β, regression coefficient; POR, prevalence odds ratio; S.E., standard error of mean; CI, confidence interval. Coding, male 1, female 2; non-smoker 1, smoker 2; code 1 is control. Data were generated from logistic regression relating POR for hypertension to seven independent variables (first column). For each test, *p*-values ≤ 0.05 indicate a statistically significant contribution of individual independent variables to the POR for hypertension.

**Table 7 toxics-11-00516-t007:** Multiple linear regression analysis of blood cadmium predictors.

Independent Variables/Factors	Log_10_([Cd]_b_ × 10^3^), µg/L					
	All Subjects		Men		Women	
	β	*p*	β	*p*	β	*p*
Age, years	−0.066	0.467	0.129	0.589	−0.101	0.314
BMI, kg/m^2^	0.105	0.217	0.246	0.266	0.115	0.241
eGFR, ml/min/1.73 m^2^	0.016	0.875	0.007	0.973	0.012	0.920
Log_10_ ([β_2_M]_s_ × 10^3^), mg/L	0.084	0.345	−0.176	0.483	0.126	0.196
Log_10_[(E_β2M_/C_cr_ × 10^3^], µg/L	0.139	0.225	0.354	0.166	0.077	0.555
Log_2_[(E_Cd_/C_cr_) × 10^5^], µg/L	0.321	<0.001	0.143	0.467	0.394	<0.001
Gender	0.110	0.275	**−**	**−**	**−**	**−**
Smoking	0.294	0.003	0.496	0.025	0.097	0.285
Diabetes	−0.175	0.060	−0.511	0.029	−0.079	0.452
Hypertension	0.143	0.078	0.164	0.473	0.181	0.050
Adjusted R^2^	0.197	<0.001	0.197	0.109	0.193	<0.001

β, standardized regression coefficient; adjusted R^2^, coefficient of determination. Coding, male 1, female 2; non-smoker 1, smoker 2; non-DM 1, DM 2; normotension 1, hypertension 2. β indicates strength of association of log_10_([Cd]_b_ × 10^3^) with independent variables (first column). Adjusted R^2^ indicates a fractional variation of log_10_([Cd]_b_ × 10^3^) explained by all independent variables. For each test, *p*-values ≤ 0.05 indicate a statistically significant contribution to the variation of log_10_([Cd]_b_ × 10^3^).

## Data Availability

All data are contained within this article.

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
