# Peer review of "Evidence Linking Cadmium Exposure and β2-Microglobulin to Increased Risk of Hypertension in Diabetes Type 2"

_toxics, 2023, doi:10.3390/toxics11060516_

Round 1
Reviewer 1 Report
The aim of this study was to test the hypothesis that Cd exposure increases blood pressure and promotes the progression of kidney disease in diabetics. The results of this survey are considered to be valuable data in Thailand, although the number of subjects is rather small. However, in terms of it ‘the progression of kidney disease’, there is not enough analysis suitable for this issue. At present, this manuscript is not acceptable for publication.
Major comment
1. It is well known that most patient of Itai-Itai disease were women. Thus, to avoid such gender difference in the effect of Cd, I strongly recommend the authors to analyze grouped according to gender.
2. (lines 227~) The aim of this study was ‘to test the hypothesis that Cd exposure increases blood pressure and promotes the progression of kidney disease in diabetics’. The results presented after ‘3.3. Determinants of Blood Cadmium’, were questionable whether these fit for this aim. In particular, it is difficult to understand how the ‘3.4 The Relationship between Serum and Urine β2M’(line 259) is relevant to the aim of this study. Furthermore, there is a lack of results assessing the association between Cd exposure and eGFR as an outcome of ‘the progression of kidney disease’. To accomplish the aim of this study, such evaluation is essential.
3. (lines 349~) Since the overall mean values are 0.59 µg/L for blood Cd and 0.84 µg/L for urinary Cd/Ccr, the ‘below’ is considered too strong. It would be better to suppress conclusive statements on B2M as well, as there are few items where significant differences were found.
Minor comments
1. (lines 177~) After this table 2, when log-transforming, the *10^3 and *10^5 in brackets can be deleted to prevent confusion in interpretation. The regression coefficients and odds ratios should remain the same as they are adjusted by the intercept on the model.
2. (lines 177:table 2) Please add DBP results for non-diabetic and diabetic participants.
3. (Table 2~) For continuous variables, indicate the units; for category variables, indicate what the results are for what (control category) in the ‘Variables’ column.
4. (line 195, Figure 1(a)) Like Figure 1(c), please add the coefficients of determination (R2) and p-values.
5. (line 195, Figure 1(b)&(d))Please indicate the tertiles used as cutoff values for three groups in the results section.
6. (lines 202-203) I consider these representative values should be shown separately for DM and non DM groups.
7. (lines 202-203) Even the values shown now are very small for 1st and 2nd groups, and relatively large in 3rd group. So please check again for calculation errors.
8. (line 228) ‘median’: Is this the median of all participants? Please indicate.
9. (lines 355-356) Please correct.
Author Response
Reviewer 1.
Comments and Suggestions
The aim of this study was to test the hypothesis that Cd exposure increases blood pressure and promotes the progression of kidney disease in diabetics. The results of this survey are considered to be valuable data in Thailand, although the number of subjects is rather small. However, in terms of it ‘the progression of kidney disease’, there is not enough analysis suitable for this issue. At present, this manuscript is not acceptable for publication.
RESPONSE: We thank the reviewer for the comments and guidance to improve our manuscript. Accordingly, below we provide point-by-point responses to each comment. Changes to the text in a manuscript are in blue.
- Major comment
It is well known that most patient of Itai-Itai disease were women. Thus, to avoid such gender difference in the effect of Cd, I strongly recommend the authors to analyze grouped according to gender.
RESPONSE
- We have clarified the study design in subsection 2.1. We have now added two supporting references (24 and 25) to the recruitment sentences (lines 81-84) The insertion made is given below.
“Previous studies suggest that the adverse effects of Cd, especially in kidneys, are more prevalent and more severe in women than men [25, 26]. Thus, more women (80.7%) were recruited to maximize the likelihood of finding an effect of Cd, when the sample size was modest (n = 176).”
[25] Nishijo, M;, Satarug, S.; Honda, R.; Tsuritani, I.; Aoshima, K. The gender differences in health effects of environmental cadmium exposure and potential mechanisms. Mo.l Cell. Biochem. 2004, 255, 87-92.
[26] Trzcinka-Ochocka, M.; Jakubowski, M.; Szymczak, W.; Janasik, B.; Brodzka, R. The effects of low environmental cadmium exposure on bone density. Environ. Res. 2010, 110(3):286-293.
- We have now provided results for the regression of SBP and DBP in men and women in a new Table 3 (lines 195-208).
- The determinants of blood cadmium in men and women are provided in a new Table 7 (lines 265-280).
- The results show a stronger effect of Cd on SBP and DBP in men than women. Our finding may help explain the results of a recent Danish cohort study of non-smokers which found an association between environmental Cd exposure and incident heart failure, especially among men (Sears et al. Urinary cadmium and incident heart failure: A case-cohort analysis among never-smokers in Denmark. Epidemiol. 2022, 33, 185-192). We have included this segment in the discussion and added this reference [ref 39].
“In men, SBP was associated strongly with both serum β2M (β = 0.415) and urinary Cd excretion (β = 0.432), while DBP was strongly associated only with urinary Cd excretion (β = 0.454). The association of the DBP in men with serum β2M was not significant (β = 0.242, p = 0.212). In comparison, there was no significant association between SBP and urinary Cd excretion women. There was, however, a moderate association of SBP with age (β = 0.310) and serum β2M (β = 0.225).”
- (lines 227~) - The aim of this study was ‘to test the hypothesis that Cd exposure increases blood pressure and promotes the progression of kidney disease in diabetics’. The results presented after ‘3.3. Determinants of Blood Cadmium’, were questionable whether these fit for this aim. In particular, it is difficult to understand how the ‘3.4 The Relationship between Serum and Urine β2M’(line 259) is relevant to the aim of this study. Furthermore, there is a lack of results assessing the association between Cd exposure and eGFR as an outcome of ‘the progression of kidney disease’. To accomplish the aim of this study, such evaluation is essential.
RESPONSE
- In the present study, we consider hypertension as a primary outcome of Cd exposure that promotes the progression of CKD, especially in diabetes. To reflect this, we have reworded the aim which now reads. “The aim of this study was to test the hypothesis that Cd exposure increases blood pressure, which in turn promotes the progression of diabetic kidney disease.”
- The subsections 3.3 and 3.4; Determinants of Blood Cadmium and the Relationship between Serum and Urinary β2M’ have been deleted.
- Our study was the first to observe an association of serum β2M with blood pressure (SBP/DBP) (Tables 2 and 3). In addition, we report that a ten-fold increase in blood Cd was associated with 1.72-fold increase in risk of hypertension (Table 6).
- The Inverse association of eGFR with serum β2M and age has now been provided in a new Table 5.
- (lines 349~) - Since the overall mean values are 0.59 µg/L for blood Cd and 0.84 µg/L for urinary Cd/Ccr, the ‘below’ is considered too strong. It would be better to suppress conclusive statements on B2M as well, as there are few items where significant differences were found.
RESPONSE
- Considering the new findings derived from the additional data analysis, that the Reviewer suggested, we have rewritten the Conclusion which is provided below.
“This study shows that environmental exposure to low levels of Cd adversely affect blood pressure and GFR in both non-diabetic and diabetic. For the first time, an independent effect of Cd and β2M on SBP and eGFR have been demonstrated. This implies that both Cd exposure and circulating β2M are involved in the development of hypertension, and eGFR decline, especially in diabetics”.
Minor comments
- (lines 177~) After this table 2, when log-transforming, the *10^3 and *10^5 in brackets can be deleted to prevent confusion in interpretation. The regression coefficients and odds ratios should remain the same as they are adjusted by the intercept on the model.
RESPONSE
- The multiplication factors 103 and 105 have been retained for uniformity.
- (lines 177:table 2) Please add DBP results for non-diabetic and diabetic participants.
RESPONSE
- The DBP results are now provided in a new Table 3
- (Table 2~) For continuous variables, indicate the units; for category variables, indicate what the results are for what (control category) in the ‘Variables’ column.
RESPONSE
- The units of continuous variables have now been added plus coding of categorical variables. Control category is also provided.
- (line 195, Figure 1(a)) Like Figure 1(c), please add the coefficients of determination (R2) and p-values.
RESPONSE
- In the original submission, the R2 and p-values values were provided but they were not displayed due to an error in file conversion. In this revised version, we have checked they are displayed correctly.
- (line 195, Figure 1(b)&(d)) Please indicate the tertiles used as cutoff values for three groups in the results section.
RESPONSE
- The cutoff values of Cd excretion for three tertile groups have been provided (lines 249-250) as quoted below.
“The cutoff values of (ECd/Ccr)×100 for tertiles 1, 2 and 3 were ≤ 0.069, 0.070-0.177 and ≥ 0.178 µg/ L filtrate, respectively.”
- (lines 202-203) I consider these representative values should be shown separately for DM and non DM groups.
RESPONSE:
- The Mean (SD) values for (ECd/Ccr) ×100 tertiles 1, 2 and 3 provided are for all participants. The mean values shown in Figures 1b and 1c all are adjusted for covariates. They could thus be considered representative mean values for DM and non-DM subgroups. In Table 1 the mean for DM and non-DM groups are unadjusted.
- (lines 202-203) Even the values shown now are very small for 1st and 2nd groups, and relatively large in 3rd group. So please check again for calculation errors.
RESPONSE:
- No calculation errors found.
- (line 228) ‘median’: Is this the median of all participants? Please indicate.
RESPONSE
- The blood Cd median level was for all participants.
- (lines 355-356) Please correct.
RESPONSE
- The statement on “Supplemental Material” was in error and has now been removed.

Reviewer 2 Report
Authors well described the relevance between cadmium, B2M, hypertension and diabetes. It is useful information that they have positive relevances. Hypertension and diabetes are very observing life style disease recently. Also, we absorbed Cd through various food; and because the biological half life of Cd is very long, it is accumulated in the body. Therefore, for the preventive medicine or preventive pharmacology. Authors gathered enough samples for the conclusion and well analyzed the data with proper statistical method.
Author Response
Reviewer 2
Comments and Suggestions
Authors well described the relevance between cadmium, B2M, hypertension and diabetes. It is useful information that they have positive relevances. Hypertension and diabetes are very observing life style disease recently. Also, we absorbed Cd through various food; and because the biological half life of Cd is very long, it is accumulated in the body. Therefore, for the preventive medicine or preventive pharmacology. Authors gathered enough samples for the conclusion and well analyzed the data with proper statistical method.
RESPONSE
- We thank the Reviewer for his/her evaluation of our manuscript. In this revised version of our paper, significant improvements have been made to all four aspects the Reviewer indicated.
- Clarification of the study design of the present study in subsection 2.1. Recruitment of Study Subjects (lines 82-83) with addition of two supporting references (ref. 25 and 26).
- The study objective has been reworded to read, “The aim of this study was to test the hypothesis that Cd exposure increases blood pressure, which in turn promotes the progression of diabetic kidney disease.”
- Results are analyzed for participants grouped by the absence/presence of diabetes, and by gender. Both hypertension and eGFR decline are analyzed as Cd exposure outcomes.
- Discussion has included a recent study a recent Danish cohort study of non-smokers which found an association between environmental Cd exposure and incident heart failure, especially among men (Sears et al. Urinary cadmium and incident heart failure: A case-cohort analysis among never-smokers in Denmark. Epidemiol. 2022, 33, 185-192) [ref 39].
- Conclusion has been rewritten.

Reviewer 3 Report
Line 92-93
Hypertension was defined as systolic blood 92 pressure ≥ 140 mmHg, or diastolic blood pressure ≥ 90 mmHg, a physician’s diagnosis, or 93 prescription of anti-hypertensive medications
How were incidental blood pressure rises – white coat hypertension excluded?
Tablea 1.
They clearly differed in the excretion of B2 microglobulin. No differences in cadmium excretion
Tablea 2:
R2 regressions are only relevant for non-diabetics. In diabetics, the regression as a whole does not meet the requirement of statistical significance.
It is worth presenting a single regression calculation for cadmium excretion alone in diabetics.
There was an imbalance in smoking between diabetics and non-diabetics. Therefore, the cadmium concentration was associated with smoking only in the non-diabetic group.
Figure 1 - no caption next to graph a
Line 337-338
It is well documented that diabetes and hypertension are 337 associated with increased risks of albuminuria, kidney disease stage 3 or 4
I suggest checking.
Line 349-350
This study shows that exposure to environmental Cd producing blood Cd levels 349 below 0.5 μg/L, and urinary Cd excretion below 0.05 μg/L of filtrate adversely affects 350 blood pressure in both non-diabetic and diabetic subjects.
Please check .
Author Response
Reviewer 3
We thank the reviewer for comments to improve our manuscript. Accordingly, below we provide point-by-point responses to each comment. Changes to the text in a manuscript are in blue.
Comments and Suggestions for Authors
- Line 92-93
Hypertension was defined as systolic blood pressure ≥ 140 mmHg, or diastolic blood pressure ≥ 90 mmHg, a physician’s diagnosis, or prescription of anti-hypertensive medications
How were incidental blood pressure rises – white coat hypertension excluded?
RESPONSE: The hypertension definition was given in error, which has now been correct to read, “Hypertension was defined as measured systolic blood pressure ≥ 140 mmHg, or diastolic blood pressure ≥ 90 mmHg (lines 95-96).
- Table 1.
They clearly differed in the excretion of B2 microglobulin. No differences in cadmium excretion
RESPONSE
We have interpreted these data to suggest that Cd exposure levels experienced by participants in the present study and the presence of diabetes were sufficient to cause impairment in tubular reabsorption of β2M.
- Table 2.
R2 regressions are only relevant for non-diabetics. In diabetics, the regression as a whole does not meet the requirement of statistical significance.
It is worth presenting a single regression calculation for cadmium excretion alone in diabetics.
There was an imbalance in smoking between diabetics and non-diabetics. Therefore, the cadmium concentration was associated with smoking only in the non-diabetic group.
RESPONSE:
- Due to modest sample size, we were unable to differentiate effectively an effect of smoking which was more prevalent in non-diabetic group than the diabetic group. Our study, however, have provided evidence for adverse effects Cd exposure from all sources in people with and without diabetes.
- Although the sample size is small, we have comprehensively analyzed data. In this revised version, eGFR has been included as outcomes of Cd exposure along with increment of SBP and DBP. The aim of this study was to test the hypothesis that Cd exposure increases blood pressure, which in turn promotes the progression of diabetic kidney disease."
- Figure 1 - no caption next to graph a
RESPONSE:
- In the original submission, the essential captions in Figure 1 were provided but they were not displayed due to an error in file conversion. In this revised version, we have checked they are displayed correctly.
- Line 337-338
It is well documented that diabetes and hypertension are associated with increased risks of albuminuria, kidney disease stage 3 or 4.
I suggest checking.
RESPONSE:
- We have now deleted the referred statement because it is a duplicate of a preceding sentence.
- Line 349-350
This study shows that exposure to environmental Cd producing blood Cd levels below 0.5 μg/L, and urinary Cd excretion below 0.05 μg/L of filtrate adversely affects blood pressure in both non-diabetic and diabetic subjects.
Please check.
RESPONSE:
- The referred blood Cd and urinary Cd excretion data have been deleted, and the conclusion has been rewritten as below.
“This study shows that environmental exposure to low levels of Cd adversely affect blood pressure and GFR in both non-diabetic and diabetic. For the first time, an independent effect of Cd and β2M on SBP and eGFR have been demonstrated. This implies that both Cd exposure and circulating β2M are involved in the development of hypertension, and eGFR decline, especially in diabetics.”

Round 2
Reviewer 1 Report
I consider the manuscript has been revised enough for publication.